



# Dried, closed-path eddy covariance method for measuring carbon dioxide flux over sea ice

Brian J. Butterworth[1], Brent G. T. Else[1]

[1]Department of Geography, University of Calgary, Calgary, T2N 1N4, Canada

*Correspondence to*: Brian J. Butterworth (brian.butterworth@ucalgary.ca)

**Abstract.** The Arctic marine environment plays an important role in the global carbon cycle. However, there remain large uncertainties in how sea ice affects air–sea fluxes of carbon dioxide ($CO_2$), partially due to disagreement between the two main methods (enclosure and eddy covariance) for measuring $CO_2$ flux ($F_{CO2}$). The enclosure method has appeared to produce more credible $F_{CO2}$ than eddy covariance (EC), but is not suited for collecting long-term, ecosystem-scale flux datasets in such remote regions. Here we describe the design and performance of an EC system to measure $F_{CO2}$ over

landfast sea ice that addresses the shortcomings of previous EC systems. The system was installed on a 10-m tower on Qikirtaarjuk Island – a small rock outcrop in Dease Strait located roughly 35 km west of Cambridge Bay, Nunavut in the Canadian Arctic Archipelago. The system incorporates recent developments in the field of air–sea gas exchange by measuring atmospheric $CO_2$ using a closed-path infrared gas analyzer (IRGA) with a dried sample airstream, thus avoiding the known water vapor issues associated with using open-path IRGAs in low-flux environments. A description of the

methods and the results from four months of continuous flux measurements from May through August 2017 are presented, highlighting the winter to summer transition from ice cover to open water. We show that the dried, closed-path EC system greatly reduces the magnitude of measured $F_{CO2}$ compared to simultaneous open-path EC measurements, and for the first time reconciles EC and enclosure flux measurements over sea ice. This novel EC installation is capable of operating year-round on solar/wind power, and therefore promises to deliver new insights into the magnitude of $CO_2$ fluxes and their

driving processes through the annual sea ice cycle.





## 1 Introduction

The global marine system plays a major role in regulating atmospheric $CO_2$, currently absorbing roughly 2 PgC from the atmosphere each year, or roughly a quarter of anthropogenic $CO_2$ emissions (Takahashi et al., 2009; Wanninkhof et al., 2013; Sitch et al., 2015). Sea ice, which covers up to 11.8% of the global ocean's surface, has important implications for the
global carbon cycle (Weeks, 2010).

Sea ice does not have the same physical properties as freshwater ice (Gosink et al., 1976). It is porous, with brine channels exchanging salt and gases between the atmosphere and the water below. Compared to terrestrial environments $CO_2$ fluxes over sea ice are small. However, there are many different types of sea ice and a large degree of uncertainty remains in the
physical processes controlling gas exchange in these regions (Miller et al., 2015). The vast size of the sea ice ecosystem means that even small exchange rates may produce important fluxes on the global scale, and therefore improved measurement techniques and increased data collection/coverage are essential to better characterize the baseline $CO_2$ exchange for sea ice regions. Such developments are also necessary for predicting of how Arctic carbon budgets will change as the current trend towards thinner, younger ice cover and reduced sea ice extent continues (Kwok, 2007; Maslanik et al.,
2007; Comiso et al., 2017).

Over the last several decades the two main approaches for measuring $F_{CO2}$ over sea ice have been the enclosure method and the eddy covariance (EC) method. The enclosure method works by measuring the change in gas concentration over time within a chamber placed on the sea ice (Miller et al., 2015). The main shortcoming with this method is that it alters the
environment which is being measured (e.g., affecting temperature, radiation, pressure gradients, wind speed, and turbulence, atmosphere-surface $CO_2$ concentration differences). Proper technique can minimize these artifacts, but even under the best conditions it is expected that they will cause some underestimation of $F_{CO2}$ (Miller et al., 2015). Additionally, the measurements are spatially and temporally limited. Measurements are confined to the region enclosed by the chamber (cm scale), making it challenging to accurately measure fluxes over whole ecosystems (m to km scale), which typically contain
heterogeneity on scales larger than the footprint of the chamber. Additionally, long-term measurements are not feasible for enclosures due to the fact that they alter the underlying environment and the degree of manual intervention they require.

The EC technique works by measuring vertical wind speed and gas concentration at high frequencies. The covariance between fluctuations in vertical wind and fluctuations in the gas concentration, averaged over a period of time, represent a
direct measurement of flux. Unlike the enclosure method it does not alter the environment in which it measures and is practical for gathering long-term continuous measurements of flux over a spatial scale that encompasses the natural heterogeneity of sea ice.



The enclosure and EC methods have not shown good agreement over sea ice, where flux magnitudes measured by EC systems have been consistently higher than those measured by enclosures (Table 1). Over landfast sea ice the enclosure method produces fluxes on the order of −5.4 to 2.2 mmol m$^{-2}$ d$^{-1}$ (Delille, 2006; Nomura et al., 2010, 2013; Sejr et al., 2011; Geilfus et al., 2012, 2014, 2015; Delille et al., 2014; Sievers et al., 2015), while EC often measures $F_{CO2}$ uptake and effluxes of several hundred mmol m$^{-2}$ d$^{-1}$ (Miller et al., 2011; Papakyriakou and Miller, 2011; Sievers et al., 2015). Fluxes of this magnitude are suspiciously high, comparing with terrestrial fluxes (Christensen et al., 2000; Lafleur et al., 2003; Sullivan et al., 2008), and fluxes over open-ocean algal blooms (Yang et al., 2016a).

The most likely explanation for the large discrepancy between methods is a failure of the open-path IRGA used in the EC systems. Overestimation of $F_{CO2}$ by open-path EC in the low-flux marine environment has been documented as far back as Broecker et al. (1986), and has since been confirmed (Miller et al., 2010; Blomquist et al., 2014; Landwehr et al., 2014). Closed-path eddy covariance systems may reduce measured $F_{CO2}$ magnitude (e.g., Sievers et al. [2015] measured a mean $F_{CO2}$ of 1.73 ± 5 mmol m$^{-2}$ d$^{-1}$ over landfast sea ice), but have been shown to be affected by the same problems as the open-path IRGAs (Blomquist et al., 2014; Landwehr et al., 2014; Butterworth and Miller, 2016b). An improved technique was developed by Miller et al. (2010) which used a closed-path IRGA and dried sample airstream. This system addressed the problems with previous EC systems by eliminating fluctuations in all variables (pressure, temperature, and $H_2O$) associated with the density correction (Webb et al., 1980). Subsequent studies have confirmed the effectiveness of this approach for measuring air–sea fluxes (Blomquist et al., 2014; Landwehr et al., 2014; Butterworth and Miller, 2016b; Bell et al., 2017).

Cavity ring-down spectroscopy (CRDS) may be suitable for measuring $F_{CO2}$ over landfast sea ice, though it has yet to be field tested. Open water results have confirmed flux detection limits for dried Picarro instruments (G1301-f; G2311-f) to be in the range needed for measuring over landfast sea ice (Blomquist et al., 2014; Yang et al., 2016b). The Los Gatos Research FGGA on the other hand (in which drying is less feasible) has flux detection limits that may not be suitable for measuring at the low flux magnitudes expected over landfast sea ice (Yang et al., 2016b), though Prytherch et al. (2017), using an undried FGGA, measured $F_{CO2}$ in the Arctic marginal ice zone that agreed with previous results (Butterworth and Miller, 2016a). Because CRDS systems are expensive and have significant power demands, systems based on closed-path IRGAs are currently more practical for making continuous flux measurements in Arctic environments.

For this study, we applied the dried, closed-path IRGA design to measure $F_{CO2}$ from a permanent tower over sea ice in the Canadian Arctic. This is the first EC system of this kind to measure $F_{CO2}$ over landfast sea ice. The benefits of a fixed tower are that it avoids the motion contamination and flow distortion associated with ship-based EC. Additionally, it is capable of collecting a long-term continuous flux dataset in one area, thus enhancing our ability to address process-level questions. Here we will present four months of flux data from the spring and summer season (May to September 2017) as the region transitioned from full ice cover to open water and describe the performance of the system. Our primary goal in this paper is





to describe the design and performance of the system, while subsequent articles will more fully explore the insights gained about $CO_2$ exchange in the sea ice environment.

## 2 Methods

### 2.1 Site Description

An eddy covariance system to measure fluxes of momentum, sensible heat, latent heat, and $CO_2$ was installed on a 10-m tower located on the northwest side of Qikirtaarjuk Island, a low-lying island (500 × 200 m) in Dease Strait, roughly 35 km west of Cambridge Bay, Nunavut (Fig. 1). Qikirtaarjuk Island is the southernmost island in a chain that extends across Dease Strait, creating active tidal straits that produce polynyas in the fall and early spring (Fig. 1). Except for the islands north of the tower, the closest being Unihitak Island 3.5 km away, the tower has unimpeded fetch on the order or 50 km from large-

angled swaths to the east and west. This ensures that much of the flux footprint represents only water and not a mix of water and land.

### 2.2 Instrument Setup

The tower was configured with an array of instruments (Fig. 2) to measure mean meteorological variables (logged as 1-minute averages on a Campbell Scientific CR1000 datalogger) and high frequency flux variables (10 Hz; CR3000

datalogger). Mean wind speed and direction were measured using a 2D propeller vane anemometer (RM Young; Marine Wind Monitor) mounted at 7.8 m above ground level (agl). Three temperature and relative humidity probes (Campbell Scientific HMP45C) were mounted at 9.6 m, 5 m, and 2.2 m agl. A net radiometer (Kipp & Zonen; CNR4) was mounted 2.8 m agl.

### 2.2.1 Wind Vector

Measurements of momentum and sensible heat fluxes were obtained using a 3D ultrasonic anemometer (CSAT3; Campbell Scientific) mounted at 9.5 m agl, oriented northwest (330°), and roughly 15 m from the water's edge. The ground level at the tower base was roughly 3 m above mean sea level (asl), making the measurement height 12.5 m asl. However, three-dimensional wind measurements show that the streamlines are directionally dependent and bend upward/downward in proportion to the island incline (a maximum of 6° from head-on winds). Therefore, wind and flux measurements were

considered representative of 9.5 m asl (similar to vertical displacement seen in ship-based measurements [Yelland et al. 2002]). For each 20-minute flux interval a double rotation was applied on the wind vector to put it into a mean streamline coordinate system where the x-axis was parallel to the mean wind ($\bar{v} = \bar{w} = 0$) (Kaimal and Finnigan, 1994).



### 2.2.2 H₂O & CO₂

Water vapor and $CO_2$ concentrations used to calculate fluxes were measured by three IRGAs. Two were open-path designs (EC150, Campbell Scientific and LI-7500, LI-COR) and one was a closed-path design (LI-7200RS, LI-COR, referred to herein as LI-7200). The EC150 was attached to the CSAT3 anemometer making its measurements collocated with the wind vector measurements. The LI-7500 was mounted 30 cm aft, 18 cm starboard, and at the same height as the CSAT3. Mixing ratios for the IRGAs were calculated from molar density, pressure, and temperature using the WPL correction (Webb et al., 1980). This was even done for the LI-7200 (using $T_{7200} = 0.8\ T_{IN} + 0.2\ T_{OUT}$ as suggested by LI-COR) because it was deemed more reliable than the LI-7200's on-the-fly calculation of mixing ratio, which inexplicably produced $F_{CO2}$ with large contributions from low frequencies.

Unlike the open-path IRGAs the LI-7200 required a pump to pull air through its cell. As pumps significantly increase power requirements, this has been one of the barriers to closed-path IRGAs being used in remote Arctic EC towers. The sample air for the LI-7200 was drawn from an inlet 5 cm aft of the CSAT3 sampling volume. Ideally, flow through the LI-7200 should be fast enough to fully flush its cell every sample (9.7 standard liters per minute [slpm] for 10 Hz sampling). However, the maximum flow rate that could consistently be achieved by our 12V DC diaphragm pump (UN814KNDC, KNF) was 7 slpm. To ensure that flow remained constant a mass flow controller (MCRW-10SLPM-D-DB9/5M, Alicat) was installed immediately upstream of the pump. It was expected that this flow rate should cause a minor loss of signal at the high frequency end of the spectrum (evaluated below).

The mounting configuration of the LI-7200 was chosen to minimize signal attenuation by tubing. The diameter of the tubing upstream of the LI-7200 was minimized (3.5 mm I.D.) to increase the Reynolds number (*Re*). With 7 slpm flow *Re* in the sample line was 2800, in the transitional zone between laminar (*Re* < 2100) and turbulent (*Re* > 4000). To mitigate the signal smoothing that could occur with non-turbulent flow, the LI-7200 was mounted near the top of the tower, reducing upstream tube length to 2.8 m.

To test for tube delay and attenuation of high frequency signal, inlet tests with $CO_2$-free air were performed regularly. Twice a day at 2:05 and 14:05 LST a normally-closed, 2-way solenoid valve installed at the base of the CSAT3 would open and inject nine 10-s pulses of N₂ directly in front of the intake tube to the LI-7200. The mean delay for $CO_2$ from 574 tests was $0.83 \pm 0.01$ s (Fig. 3). The mean time constant (defined as the time for signal to drop below 1/e strength) was $0.24 \pm 0.02$ s. This time constant (less than three samples at 10Hz) was low and suggested minimal attenuation by the tubing. High frequency loss of $F_{CO2}$ was characterized by estimating the flux lost by the open-path latent heat flux measurement after applying a low-pass filter to H₂O mixing ratio using the $CO_2$ time constant as a cutoff frequency (Goulden et al., 1996). The ratio ($G_c$) of unfiltered to filtered latent heat flux indicated an average high frequency $F_{CO2}$ loss of 1.7%, in line with previous




studies (Ibrom et al., 2007; Butterworth and Miller, 2016b). To account for this loss, while reducing the variability in $G_c$, a linear regression between $G_c$ and $U_{10n}$ was calculated ($G_c = 0.002\ U_{10n} + 1.0024$) and used to compute a multiplier for each flux interval based on the wind speed.

### 2.2.3 Sea Ice

Images of sea ice were captured by a camera (Hero4, GoPro) mounted at the top of the tower. An intervalometer was installed to take a picture once an hour, indefinitely. These images were to be used to determine sea ice concentration (SIC) and melt pond fraction. However, the external battery packs for the camera failed late May 2017, and no images were collected by the camera until the issue was fixed mid-July 2017. Because of this, we relied on several other methods for estimating SIC. The AMSR2 passive microwave SIC product (daily, 3.125 km) by University of Bremen (Spreen et al., 2008) was used to provide a picture of the seasonal ice breakup of the area. The ice concentration from the three grid cells nearest the tower were averaged. In addition to this product a variety of remotely-sensed (Landsat-8 and MODIS) and in situ images were collected. In situ photographs were taken during site visits (four helicopter trips in June and July) and from a motion sensor-equipped trail camera (installed to identify wildlife interactions with the installation, e.g. Fig. S1, but which was frequently set off by environmental conditions). Comparisons between the AMSR2 SIC product and photographs confirms that it was generally accurate within the Dease Strait region (Fig. 4), with the exception that it could not resolve melt ponds as different from open water. This meant that during the melt pond season (June) the product underestimated SIC due to the presence of overlying water. Combining the photographs with the SIC product enabled estimates of melt pond fraction. Additionally, the AMSR2 product continued to measure SIC in mid-July, after images revealed full open water in front of the tower (Fig. 4). This discrepancy was due to the fact the AMSR2 product had a footprint that extended beyond the immediate area in front of the tower (which turns to open water more quickly than the rest of the region).

### 2.2.4 Power

Power limitations often exert a large influence on experimental design in Arctic field studies. With a closed-path IRGA and airstream drying this study required two pumps and a mass flow controller, and needed roughly 4x the power required by an open-path system. With no external power to draw from, all power needed to be generated on site. Three 150W solar panels (EWS-150P-36, Enerwatt) and one 12V DC wind turbine (AIR Breeze, Primus) were used to generate power that was stored in a battery bank of five 92AH AGM batteries. The battery bank was housed in a large Pelican case, and included charge controllers and circuit breakers for both the solar panels (30A 12VDC EWC-30, Enerwatt) and turbine (Wind Control Panel, Primus). The solar panels were arranged in a triangular formation to collect solar radiation at different times throughout the long summer days. The turbine was used as supplemental power to enable power generation when solar panels were not active (night and winter). To conserve power the flux system was design to shut off the more power-hungry equipment (gas analyzers, mass flow controller, DC pumps, and sonic anemometer) when voltage in the battery bank dropped below 11.8 V,


and turn them back on again when voltage rose above 12.3 V. The system was fully operational 99.2% of the time during the study period. A schematic diagram (Fig. S2) of the power system is included in the Supplemental Material.

### 2.2.5 Drying

Like previous dried airstream systems our system used a moisture exchanger (Nafion PD-200T-12MPS, Permapure) to dry the sample air. However, instead of a using a zero-air generator for purging water vapor from the counterflow (which would have required AC power and compressed air), a desiccant (Du-Cal Drierite) was used. Air was pumped (UN814KNDC, KNF) in a closed loop through a large cylindrical tank containing 50 lbs. of desiccant, to the moisture exchanger, and back to the tank. The advantage of using a closed loop design is that the only moisture exposed to the desiccant was that which had passed through the moisture exchanger membrane. This, along with the large mass of desiccant, meant that the replacement of desiccant was required only once every 40–60 days, depending on ambient humidity. The need for desiccant replacement was determined with a small amount (1 lbs.) of Indicating Drierite (which changes color when exhausted) placed by a glass window on the tank.

### 2.3 Data Processing

Fluxes of momentum ($\tau$, N m$^{-2}$), sensible heat ($H_S$, W m$^{-2}$), latent heat ($H_L$, W m$^{-2}$), and CO$_2$ ($F_{CO2}$, mmol m$^{-2}$ d$^{-1}$) were calculated for 20-minute intervals as

$$\tau = \overline{\rho_a}\sqrt{\overline{u'w'}^2 + \overline{v'w'}^2}\,, \tag{1}$$

$$H_s = \overline{\rho_a}c_p\overline{w'T'}, \tag{2}$$

$$H_L = \overline{\rho_a}L_v\overline{w'q'}, \tag{3}$$

$$F_{CO2} = \overline{\rho_a}\,\overline{w'c'}, \tag{4}$$

where $u$, $v$, and $w$ (m s$^{-1}$) are the along-wind, cross-wind, and vertical wind components, respectively, $\overline{\rho_a}$ (mol m$^{-3}$) is the mean dry air density, $c_p$ (J kg$^{-1}$ K$^{-1}$) is specific heat capacity of air, $T$ (K) is dry air temperature, $L_v$ (J kg$^{-1}$) is latent heat of vaporization, $q$ (kg kg$^{-1}$) is specific humidity, $c$ is CO$_2$ mixing ratio (μmol mol$^{-1}$), primes indicate fluctuations about the mean, and the overbar corresponds to the time average. The dry air temperature was calculated from the sonic temperature after correction for the effect of water vapor on air density and speed of sound (Schotanus et al., 1983).

The mean wind speed from the sonic anemometer was adjusted to neutral stability at 10 m height using a semi-logarithmic wind profile and assuming a constant flux layer according to

$$U_{10n} = (u_*/\kappa)\ln(10/z_0), \tag{5}$$





where $u_* = (\tau / \rho)^{1/2}$ is friction velocity (m s$^{-1}$) measured by CSAT3, $\kappa$ is the von Karman constant of 0.4, $z$ is measurement height, and $z_0$ is roughness length (m) calculated as

$$z_0 = z \exp\left[-\frac{\kappa \bar{U}(z)}{u_*} - \psi_m\left(\frac{z}{L}\right)\right], \tag{6}$$

where $\psi_m$ represents the stability function of Paulson (1970) for unstable stratification and Grachev et al. (2007) for stable
stratification, both functions of $z/L$, where $z$ is measurement height and $L$ is Obukhov length, calculated as

$$L = \frac{T}{\kappa g}\left(\frac{u_*^3}{\overline{w'T'}+\frac{0.61\,T}{1+0.61\,Q}\overline{w'q'}}\right), \tag{7}$$

where g is the acceleration due to gravity, $T$ is air temperature, $Q$ is specific humidity, and $\overline{w'T'}$ and $\overline{w'q'}$ are turbulent fluxes of temperature and water vapor (Andreas et al., 2010).

Quality control criteria were used to select intervals that passed the underlying assumptions of eddy covariance. First, wind directions of $-150°$ to $150°$ (relative to the anemometer) were selected to eliminate winds from aft that were affected by flow distortion from the instruments and tower frame. Winds from aft were also unwanted as they included the island land surface in their flux footprint. A second quality control criterion selected for intervals that exhibited stationarity following

$$RN_{cov} = \left|\frac{\overline{(w'x')}_5 - \overline{(w'x')}_{20}}{\overline{(w'x')}_{20}}\right| \le 0.3, \tag{8}$$

where $\overline{(w'x')}_5$ represents the mean of the four 5-minute turbulent flux subintervals and $\overline{(w'x')}_{20}$ represents the turbulent flux of the whole 20-minute interval (Blomquist et al., 2014). The purpose of this criterion is to identify and remove intervals in which large scale phenomena (e.g., mesoscale motions), outside the frequency range of turbulent fluxes, are contributing to the measured flux.

### 2.3.1 Low Frequency Contribution

One issue that was encountered during flux processing was unexpected low frequency (between $10^{-3}$ to $10^{-2}$) contribution to $F_{CO_2}$, separated from contributions from the typical frequency range for turbulent fluxes by a distinct spectral gap (Fig. 5a). The spectral gap suggests that the low frequency contributions were the result of larger-scale motions (e.g., advection) and not representative of locally meaningful fluxes. To remove this from the flux measurements a high-pass filter (1$^{st}$ order Butterworth filter with cutoff frequency of 0.005 Hz centered on the trough in the spectral gap) was applied to the $CO_2$
mixing ratio prior to calculating $F_{CO_2}$. This reduced $F_{CO_2}$ magnitude by an average of 15.8% (or 0.6 mmol m$^{-2}$ d$^{-1}$; Fig. 5a). The choice to filter had to balance the need to remove spurious flux with the possible removal of real flux operating at lower frequencies (Sakai et al., 2001; Finnigan et al., 2003). To assess the loss of real flux we applied the same high-pass filter to $T$ and calculated $H_S$. Filtering caused an average flux loss of 3.5%; represented in Fig. 5b as the area between the unfiltered



and filtered $H_S$ cospectra. This real flux loss is substantially less that that lost by filtering $F_{CO2}$, which indicates that the filtering was appropriate in this instance. Future investigations into the processes affecting $F_{CO2}$ (e.g., melt pond fraction, sea ice concentration) would benefit from a more subjective review of cospectra for individual flux intervals. Because $F_{CO2}$ is presented more broadly in this paper, that level of scrutiny is not warranted here.

**3 Results**

**3.1 Meteorology**

The period reported in this study ranges from May 4 to September 1, 2017, encompassing the transition from full ice coverage to fully open water. From May 4 through May 25 the study area was characterized by snow-covered by sea ice. With air temperatures well below 0 C, this period represents winter ice conditions. From May 25 to June 25 melt ponds

began to form. Comparing the AMSR2 SIC product to the in situ photographs (which show no open water) suggests that the melt ponds during this period ranged between from between 0 to 50% of the surface area. Following this period there was an ice breakup period (June 25 to July 7) which exhibited both ice and open water. This breakup initially occurred directly in front (north) of the tower, creating a polynya that was probably caused by tidal currents in the strait funneled between the islands (Fig. 1). By the end of the breakup period the area was ice free for the remainder of the summer season.

The meteorology of the study area through this period shows the strong seasonal shift. The temperature over this period rose from its minimum of −24 C on May 5, to its maximum of 18 C on August 13, with over half of the time in-between being within ±5° of 0 C (Fig. 6b). Incoming solar radiation exhibited strong diurnal trends, with over 75% of days experiencing peak daytime values greater than 500 W m$^{-2}$, and nighttime minimum values near zero (Fig. 6a). These oscillations did not

result in large diurnal temperature swings. Due to the low temperatures, the relative humidity was typically high, with a mean of 86% (Fig 6c). Actual water vapor content of the air was lowest in May, with a mean of 3.3 ppt, followed by a mean of 7.7 ppt from June to September (Fig 6h). The $CO_2$ mixing ratio was roughly 410 ppm at the start of May and decreased to 403 ppm by the end of August, indicative of the seasonal trend that occurs when plant biomass consumes atmospheric $CO_2$ in the boreal growing season (Fig 6g). Wind speed was moderate, ranging between 0 and 16.4 m s$^{-1}$ with a mean of 5.6 ± 2.7

m s$^{-1}$. The winds exhibited no distinct change in magnitude over the course of the season (Fig. 6e), and were most commonly out of the southeast (105° to 135°) and the west southwest (235° to 275°)(Fig. 6f, Fig. 7). These directions were generally advantageous due to the large fetch in the east and west directions, the only caveat being the very small island 1.5 km southeast of the tower.





### 3.2 Air–sea fluxes

Time series of $\tau$, $H_S$, $H_L$, and $F_{CO2}$ over the course of this study are shown in Fig. 8. The $\tau$ ranged from 0 to 0.41 N m$^{-2}$, with a mean of 0.05 ± 0.05 N m$^{-2}$ (Fig. 8a). The range in $\tau$ was similar through all different surface conditions, being most strongly influenced by wind speed. The $H_S$ showed a diurnal trend (increasing during the day, decreasing at night)

throughout much of the study period. The sea surface type also played a role in mean $H_S$, with the flux during full ice cover generally upward, then, during ice breakup and early summer – downward, and in the open water at the end of the summer – upward again. $H_L$ acted similarly to $H_S$, with upward fluxes early in the season (full ice and melt ponds), transitioning to downward fluxes during ice breakup, and upward fluxes under full open water conditions.

### 3.3 CO$_2$ flux

$F_{CO2}$ measured by the dried, closed-path system was low during periods with sea ice cover. During full ice cover $F_{CO2}$ hovered around zero with a mean of −0.03 ± 1.21 mmol m$^{-2}$ d$^{-1}$. During melt season $F_{CO2}$ was slightly negative (i.e., downward) with a mean of −0.34 ± 2.04 mmol m$^{-2}$ d$^{-1}$. During ice breakup, when the surface was mixed ice and water, $F_{CO2}$ was downward (mean of −2.9 ± 4.9 mmol m$^{-2}$ d$^{-1}$). By the full open water period in August, $F_{CO2}$ had switched direction and the water was outgassing CO$_2$ to the atmosphere (mean of 3.8 ± 4.7 mmol m$^{-2}$ d$^{-1}$).

$F_{CO2}$ calculated from the different IRGAs are shown in Fig. 9. Both open-path IRGAs yielded flux values with magnitudes much larger than those from the dried LI-7200, sometimes orders of magnitude larger. For example, during full ice conditions, when the dried LI-7200 measured near-zero $F_{CO2}$, the LI-7500 and EC150 had means of −22 ± 58 and −32 ± 103 mmol m$^{-2}$ d$^{-1}$, respectively.

Because comparisons of $F_{CO2}$ from different gas analyzers do not have a dependent variable we used Pearson's correlation coefficient (r) used to describe the linear correlations between the two variables (Goodrich et al., 2016). Comparisons of $F_{CO2}$ from the closed-path LI-7200 against $F_{CO2}$ from the two open-path IRGAs showed no correlation (Fig. 10a,b), with r = 0.04 and r = 0.15 for the LI-7500 and EC150, respectively. An orthogonal regression of $F_{CO2}$ (LI-7500) against $F_{CO2}$ (EC150)

yielded a fit closer to 1-to-1 (Fig 10c), but with more scatter and an r = 0.44. On the other hand, the regression for $H_L$ (LI-7500) against $H_L$ (EC150) showed a distinct 1-to-1 relationship (Fig 10d). The correlation coefficient for this case was 0.93, thus showing a strong linear relationship. This suggests that the open path IRGAs are better suited to measuring $H_L$ than $F_{CO2}$ in this environment.

$F_{CO2}$ from all three IRGAs were also compared against heat fluxes, $H_L$ and $H_S$ (Fig. 11). Negative relationships were found between $F_{CO2}$ from the open-path IRGAs and heat fluxes. $F_{CO2}$ from the dried, closed-path IRGA showed no relationship with $H_L$ or $H_S$.



## 4 Discussion

### 4.1 Sea ice flux comparisons

During the spring season prior to ice breakup the dried, closed-path EC system measured $F_{CO2}$ of $-0.25 \pm 1.75$ mmol m$^{-2}$ d$^{-1}$. When only considering sea ice conditions prior to melt pond formation $F_{CO2}$ was $-0.03 \pm 1.21$ mmol m$^{-2}$ d$^{-1}$. These

measurements are within the range measured by previous enclosure measurements, which taken together span from $-5.4$ to 2.2 mmol m$^{-2}$ d$^{-1}$ (Table 1; [Delille, 2006; Nomura et al., 2010, 2013; Sejr et al., 2011; Geilfus et al., 2012, 2014, 2015; Delille et al., 2014; Sievers et al., 2015]). The measurements also exhibit a sharp divergence from previous open-path EC $F_{CO2}$ measurements, which at 10s to 100s of mmol m$^{-2}$ d$^{-1}$ are several orders of magnitude larger (Semiletov et al., 2004; Zemmelink et al., 2006; Else et al., 2011; Miller et al., 2011; Papakyriakou and Miller, 2011). Unlike the dried, closed-path

system our open-path systems installed at our site did measure $F_{CO2}$ with similar magnitudes to these previous open-path EC studies, with mean values of $-22 \pm 58$ (LI-7500) and $-32 \pm 103$ (EC150) mmol m$^{-2}$ d$^{-1}$.

This disagreement between simultaneous open-path and dried, closed-path at our site suggests that the reason for discrepancies between previous chamber and open-path EC measurements were not the result of different scales of

measurement, but rather problems with the ability of open-path EC to resolve fluxes. This is further demonstrated by the poor agreement between the two open-path $F_{CO2}$ results (Figure 10c).

### 4.2 Heat fluxes

Previous undried EC studies over the open ocean have found relationships between heat fluxes ($H_S$ and $H_L$) and $F_{CO2}$ (Landwehr et al., 2014; Sievers et al., 2015). However, in these instances the magnitude of measured $F_{CO2}$ at high latent heat

fluxes exceed values calculated from bulk $F_{CO2}$ formula. In their comparison of dried and undried closed-path EC systems Landwehr et al. (2014) concluded that such relationships represented bias, and did not result from real physical phenomena. They also found that the degree of bias was different for each individual IRGA instrument. While our OP IRGAs found a relationship between heat fluxes and $F_{CO2}$ our dried, closed-path system did not (Fig. 11). This supports the finding that these relationships represent bias, further evidence that open-path IRGAs do not fully remove the effects of $H_S$ and $H_L$ in the

density correction and/or the instruments' built-in water vapor corrections.

### 4.3 EC150

To our knowledge, this study is the first published test of the EC150 against the LI7500 in a marine environment. The two instruments produced similar $H_L$ (Fig. 10d), showing that the both are capable instruments for measuring water vapor flux. When it came to $F_{CO2}$ the EC150 values diverged from the dried, closed-path system to a similar degree as the LI-7500 (Fig.

10b). Like the LI-7500 the EC150 also showed strong negative relationship between $F_{CO2}$ and both $H_S$ and $H_L$ (Fig. 11). These findings suggest that the EC150 is affected by the same problems that affect the LI-7500.





However, $F_{CO2}$ comparison between the EC150 and LI-7500 did not show strong agreement (Fig. 10c). It is possible that the disagreement between the two stems from differences in their design (e.g., the EC150 is not necessarily affected by the same instrument-induced $H_S$ as the LI-7500 [Burba et al. 2008], and presumably has a different set of equations accounting for

water vapor cross sensitivity). But the overall spurious, high magnitudes for $F_{CO2}$ appear to stem from problems inherent to the open-path design. The EC150, like the LI-7500, appears to be more appropriate for use in regions with larger magnitude $F_{CO2}$.

### 4.4 Gas transfer velocity

While measuring $F_{CO2}$ in the same range as chamber measurements shows that the method ameliorates problems associated

with open-path systems, it alone is not a full accounting of measurement quality. To further assess the performance of the flux system we compared our open water results against those from previous studies. To do this we calculated gas transfer velocity, a coefficient which describes the efficiency of gas transport across the air–sea interface. Gas transfer velocity is a more effective comparison than $F_{CO2}$ because it provides more context. It was calculated by setting our measured flux equal to the bulk $CO_2$ flux formula ($F_{CO2} = k\, s\, [pCO_{2w} - pCO_{2atm}]$), and rearranging the equation to form

$$k = \frac{F_{CO2}}{s\,(pCO_{2w} - pCO_{2atm})},\tag{9}$$

where $k$ is gas transfer velocity, $s$ is solubility of $CO_2$ in seawater, $pCO_{2w}$ and $pCO_{2atm}$ are the partial pressure of $CO_2$ in water and the atmosphere, respectively (Wanninkhof and McGillis, 1999). While $F_{CO2}$ and $pCO_{2atm}$ were continuously measured by the EC system, $s$ and $pCO_{2w}$ were not. They were however, measured aboard the research vessel (RV) *Martin Bergmann*, which made several courses past the island in August 2017. For the flux intervals that aligned temporally with

these passes (all fully open water) we calculated $k_{660}$ ($k$ adjusted to a Schmidt number [$Sc$] of 660). The $k_{660}$ values plotted against $U_{10n}$ (Fig. 12) showed good agreement with previous parameterizations of $k_{660}$ (Wanninkhof, 1992, 2014). Because the (RV) *Martin Bergmann* data showed that $pCO_{2w}$ had high temporal and spatial variability in this region, and because we only have three data points, this result should not be interpreted as a new functional form to the $k$ versus $U_{10n}$ relationship. What it does indicate, is that the dried, closed-path flux system is able to resolve $F_{CO2}$ within an expected range, based on

previous results. Conversely, $k_{660}$ from the open-path IRGAs were not similar to previous findings. For these three intervals, the mean $k_{660}$ was 247 cm hr$^{-1}$ for the EC150 and $-28$ cm hr$^{-1}$ for the LI-7500 (where negative represents counter gradient flux). This provides additional evidence that the open-path IRGAs are not capable of resolving $CO_2$ fluxes in this environment.

The quality of the flux measurement was also assessed by comparing previous years' measurements of $pCO_{2w}$ to estimated $pCO_{2w}$ (using $F_{CO2}$) from our study period. Measurements of $pCO_{2w}$ were collected aboard the Canadian Coast Guard (CCGS) Ice Breaker *Amundsen* near Qikirtaarjuk Island during five previous summers (2010, 2011, 2014, 2015, 2016).



August measurements of pCO$_{2w}$ (collected within a 10 km radius of the tower) ranged from 360 to 469 µatm with a mean of 407 ± 34 µatm. Estimates of pCO$_{2w}$ from our study period were calculated by rearranging Eq. (9) so that $pCO_{2w} = F_{CO2} \, k^{-1}s^{-1} + pCO_{2atm}$. As stated above, pCO$_{2atm}$ and F$_{CO2}$ were measured by the tower. Gas transfer velocity was obtained using the Wanninkhof (2014) parameterization with measured $U_{10n}$, plus the mean Schmidt number (*Sc*) from the RV *Martin*

*Bergmann* dataset (*Sc* = 1150). Solubility was also estimated as the mean value from the RV *Martin Bergmann* data (*s* = 48 mol m$^{-3}$atm$^{-1}$).

Using F$_{CO2}$ from the dried, closed-path IRGA yielded pCO$_{2w}$ estimates ranging of 347 to 481 µatm (10$^{th}$ to 90$^{th}$ percentile), and a median value of 407 µatm, over the course of summer 2017. This matched reasonably well with the range identified by

the CCGS *Amundsen* measurements. Comparatively, the pCO$_{2w}$ values from the open-path IRGAs were not as tightly clustered around a central value (Fig. 13). The fluxes from the LI-7500 and EC150 led to pCO$_{2w}$ estimates ranging from −1255 to 1101 µatm and −1252 to 906 µatm, respectively. These values are far beyond the magnitude observed in this region (and with no physical basis in the case of negative values); further evidence that the open-path IRGAs are not capable of providing accurate F$_{CO2}$ measurements in the marine environment.

**4.5 Drying**

The drying system worked as desired, drying the sample air to dewpoint temperatures of −27.5 ± 7.6 C, compared to the LI-7500 which measured average ambient dewpoint of 6.0 ± 7.6 C. Perhaps more importantly it reduced fluctuations in water vapor, reducing the standard deviation in H$_2$O mixing ratio over an order of magnitude from 0.1 ± 0.2 mmol mol$^{-1}$ to 0.007 ± 0.008 mmol mol$^{-1}$. This resulted in reducing the standard deviation of the CO$_2$ mixing ratio from 0.7 ± 1.8 µmol mol$^{-1}$ to 0.1

± 0.1 µmol mol$^{-1}$. This matches the LI-7200's specification for RMS noise at a sampling rate of 10 Hz, showing that preconditioning the sample air completely removed the influence of other variables on the CO$_2$ measurement.

As the comparisons between the LI-7200 and LI-7500/EC150 show, drying the air did seem helpful in producing F$_{CO2}$ that are more in line with expected values. Interestingly however, there were two occasions where the desiccant's capacity ran

out and the closed-path IRGA was receiving sample air with near-ambient water vapor content (Figure 6h). During those periods of time, the LI-7200 still experienced standard deviations in H$_2$O mixing ratio 3.5 times lower than the open-path LI-7500. Additionally, there was no increase in the variance of F$_{CO2}$ from the LI-7200 during these periods, and F$_{CO2}$ magnitudes did not increase to open-path levels. This suggests that even without a dry counterflow, the Nafion drier still improves F$_{CO2}$ measurements to an acceptable level by reducing fluctuations in water vapor. This makes sense because

spikes of moister or drier air will still exchange H$_2$O with a counterflow that is at mean ambient humidity. Without a parallel, undried LI7200 it impossible to quantify how much of the smoothing is from the Nafion compared to the natural 'stickiness' of H$_2$O on tube walls. However, the impact of H$_2$O smoothing from tube walls alone was tested in Butterworth and Miller




(2016b) with an undried LI7200. It was found that tubing did not fully remove the influence of water vapor on the $CO_2$ flux, and instead showed up as spurious low frequency contribution to the flux, which was visible in the flux cospectra. In this study, when the desiccant was exhausted, the cospectra did not indicate interference from water vapor, suggesting that that the Nafion played a critical role. This finding may have important implications for the design of future systems (i.e., a system could be designed that uses a Nafion and counterflow, but without a dry air source), and should be investigated further.

## 4.6 Future work

In April 2018 as part of the Polar Knowledge Canada funded CAT-TRAIN project in collaboration with the Arctic Research Foundation, a mobile power station/research lab was installed at the site. This new infrastructure will be used to increase the functionality of the system. Specific additions that are being considered are incorporating waterside $pCO_{2w}$ measurements, to be used to calculate gas transfer velocity continuously through an annual cycle. This would be particularly useful considering the $pCO_{2w}$ measurements from the CCGS *Amundsen* and the RV *Martin Bergmann*, which showed that this region often has $\Delta pCO_2$ of sufficient magnitude ($> 40$ µatm) to measure accurate gas transfer velocities.

Additionally, for data redundancy we plan to install a second closed-path greenhouse gas analyzer (CRDS) capable of measuring $CO_2$, $H_2O$, and methane (GGA-FGGA, Los Gatos Research), which we which have only deployed on ships due to the large power consumption. Next spring a planned intercomparison study taking place in Cambridge Bay will add simultaneous enclosure measurements to verify agreement between the two methods. Lastly, the system will be used (in forthcoming papers) to investigate annual gas exchange cycles and process-level questions, including the processes affecting $F_{CO2}$ during spring melt and autumn freeze-up.

## 5 Conclusions

With its vast spatial extent, sea ice has the potential to play an important role in the global $CO_2$ cycle. Unfortunately, there has been significant confusion around the importance of that role, largely because the community studying sea ice gas fluxes has been unable to reconcile large fluxes measured by eddy covariance with significantly smaller fluxes measured by enclosure methods (Table 1). This problem is analogous to the problem faced by researchers studying open water gas exchange, where for several decades EC measurements could not be reconciled with tracer-based measurements (e.g., Broecker et al. [1986]). The open water problem was eventually resolved by using closed-path EC systems with a dried sample airstream (Miller et al., 2010), and EC measurements are now better aligned with other techniques.

The dried, closed-path IRGA method has previously been applied to the marginal ice zone (i.e., open water and drifting ice) where the open water likely dominates the $CO_2$ flux signal (Butterworth and Miller, 2016a). In this study, we have for the



first time applied sample drying techniques to an installation capable of measuring $CO_2$ fluxes throughout an annual sea ice cycle. This allowed for measurements over many different surface conditions, including landfast sea ice, ice break-up, and open water. Fluxes measured during the open water season matched well with existing gas transfer parameterizations, lending credibility to the method. During the ice-covered season, this new measurement system closed the gap between EC

and enclosure methods, producing $F_{CO2}$ with magnitudes in the range found by enclosure studies. This finding suggests that modeling or upscaling studies aiming to estimate the global $CO_2$ exchange associated with landfast sea ice should focus on the smaller range of $CO_2$ fluxes published by enclosure studies, at least until the EC method presented in this paper can be applied to more sea ice environments.

The dried, closed-path EC method presented here represents a significant advancement from previous attempts to measure $F_{CO2}$ over sea ice. We showed that incorporating the additional system complexity is feasible, even in remote polar locations, by presenting an effective approach for drying under low power requirements. The improved system can obtain long-term, continuous $F_{CO2}$ measurements over larger spatial scales than is possible with enclosures and opens potential avenues for new research, including a greater scrutiny of the ecosystem-scale processes affecting $CO_2$ fluxes sea ice regions.

**Data availability**

The flux data from this study are not currently being placed in a data repository. We plan to publish the dataset as one full year of flux data with an upcoming paper highlighting the annual flux cycle.

**Author contribution**

Brian Butterworth and Brent Else designed and installed the flux system. Brent Else secured the grant funding for the
research activities, and organized field logistics. Brian Butterworth processed and analyzed the flux data. Brian Butterworth prepared the manuscript with contributions from Brent Else.

**Competing interests**

The authors declare that they have no conflict of interest.

**Acknowledgements**

We wish to thank the many students and technicians who helped install and maintain the eddy covariance tower, in particular: Shawn Marriott, Patrick Duke, Angulalik Pederson, Jasmine Tiktalek, Laura Dalman, and Vishnu Nandan. The deployment of this tower in such a challenging location would not have been possible without the excellent logistical support



provided by Polar Knowledge Canada, the Arctic Research Foundation, and the Polar Continental Shelf Program. Financial

support was provided by the Marine Environmental Observation Prediction and Response (MEOPAR) Network of Centres

of Excellence, Polar Knowledge Canada, the Natural Sciences and Engineering Research Council of Canada (NSERC), the

Canada Foundation for Innovation John R. Evans Leaders Fund, the Nunavut Arctic College, Irving Shipbuilding Inc., and

5   the University of Calgary. A special thanks to the Ekaluktutiak Hunters & Trappers Organization for their continued support

of our projects, and for the expert assistance provided by their guides. This paper is a contribution to SCOR Working Group

152 – Measuring Essential Climate Variables in Sea Ice (ECV-Ice).

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

**Table 1. Eddy covariance (open-path IRGAs) and enclosure measurements of $F_{CO_2}$ (mmol m$^{-2}$ d$^{-1}$) over landfast sea ice.**

| | Study | Range | |
|---|---|---|---|
| | | Low | High |
| Eddy Covariance | Semiletov et al., 2004 | −38.6 | −19.5 |
| | Zemmelink et al., 2006[a] | −18.2 | −4.5 |
| | Semiletov et al., 2007 | −3.5 | −0.4 |
| | Else et al., 2011[a] | 20 | 36 |
| | Miller et al., 2011[a] | −60.5 | 70.3 |
| | Papakyriakou and Miller, 2011[a] | −259.2 | 86.4 |
| | Sievers et al., 2015 | −110 | 295 |
| Enclosure | Delille, 2006 | −4 | 2 |
| | Nomura et al., 2010 | −1.0 | 0.7 |
| | Sejr et al., 2011 | 0 | 1.1 |
| | Geilfus et al., 2012 | −2.65 | 2.1 |
| | Nomura et al., 2013 | −4.0 | 0.5 |
| | Geilfus et al., 2014 | −2.9 | 0.3 |
| | Delille et al., 2014 | −5.2 | 1.9 |
| | Geilfus et al., 2015 | −5.4 | −0.04 |
| | Sievers et al., 2015 | 0.9 | 2.2 |

[a] Results were reported as a range of daily averages (5-hr average in the case of Papakyriakou and Miller 2011). Full measurement ranges are necessarily larger, though unreported.





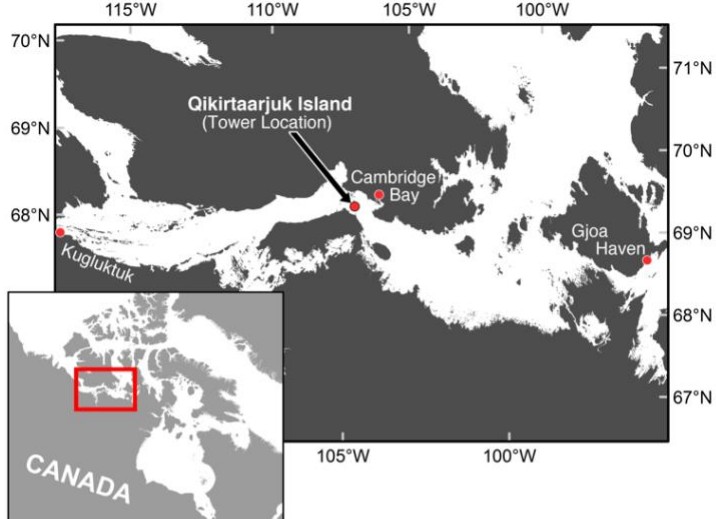

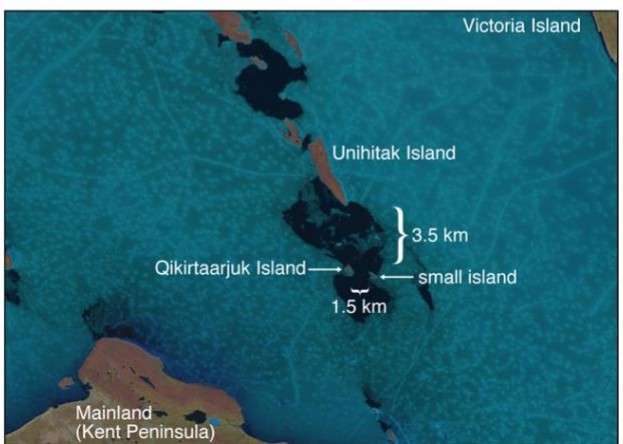

**Figure 1. Map (top) showing the location of Qikirtaarjuk Island, 35 km west of Cambridge Bay, Nunavut. Satellite image (bottom) of Qikirtaarjuk Island (June 28, 2017), showing polynya development in the tidal straits. Landsat-8 image courtesy of the U.S.**
5 **Geological Survey.**





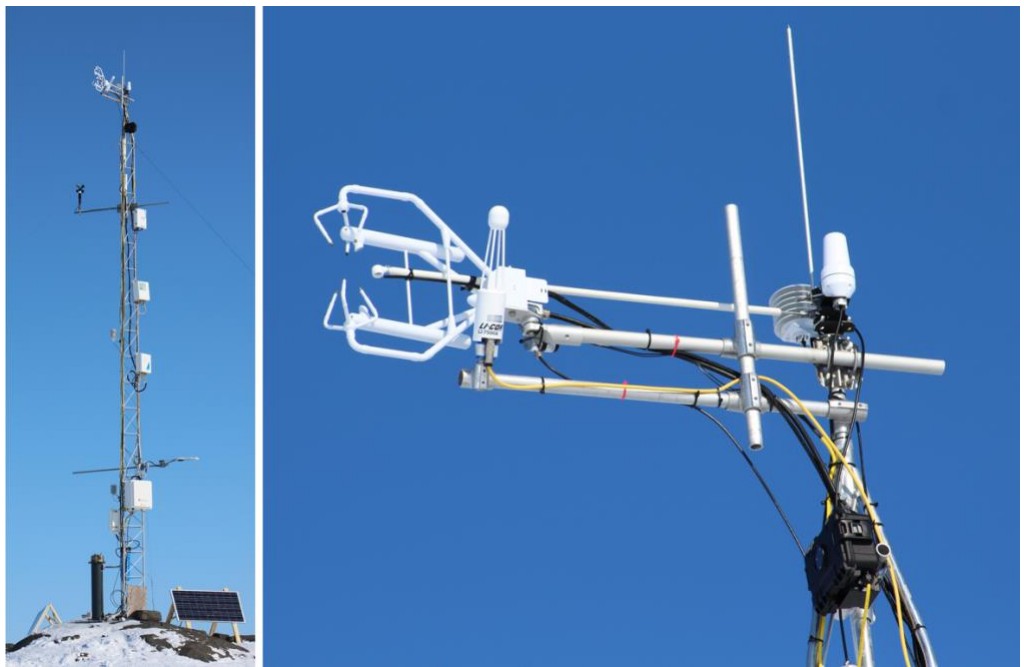

**Figure 2. Photograph of the tower (left) and the flux instruments mounted at the top of the tower (right).**

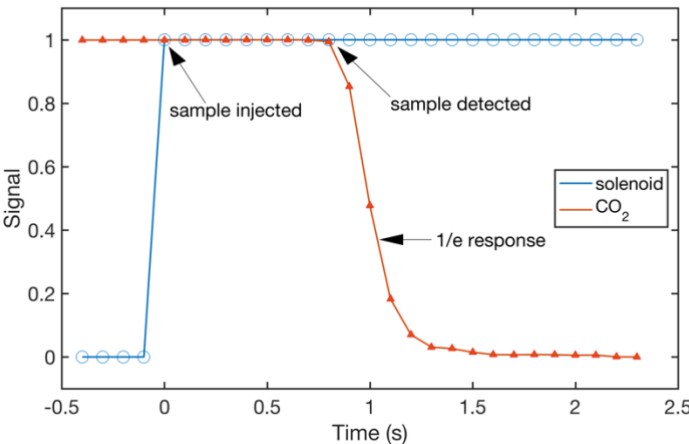

5    **Figure 3. Mean tube delay obtained from 574 inlet tests. Blue circles show the state of the solenoid valve (0 = closed; 1 = opened) responsible for releasing compressed N₂ directly in front of the sample tube inlet to the closed-path IRGA. The red triangles represent the decay in the CO₂ mixing ratio from its pre-test steady state (1) to its settled value during test (0).**



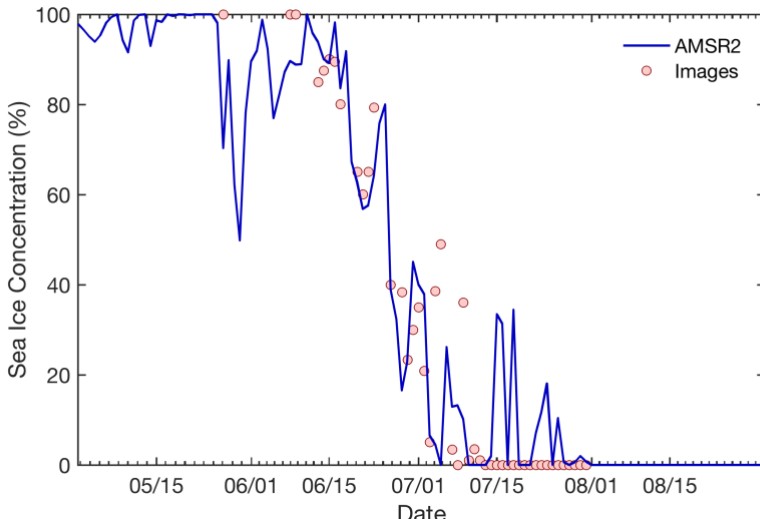

**Figure 4. Sea ice concentration from the AMSR2 SIC product and the mean daily average from satellite and in situ images.**



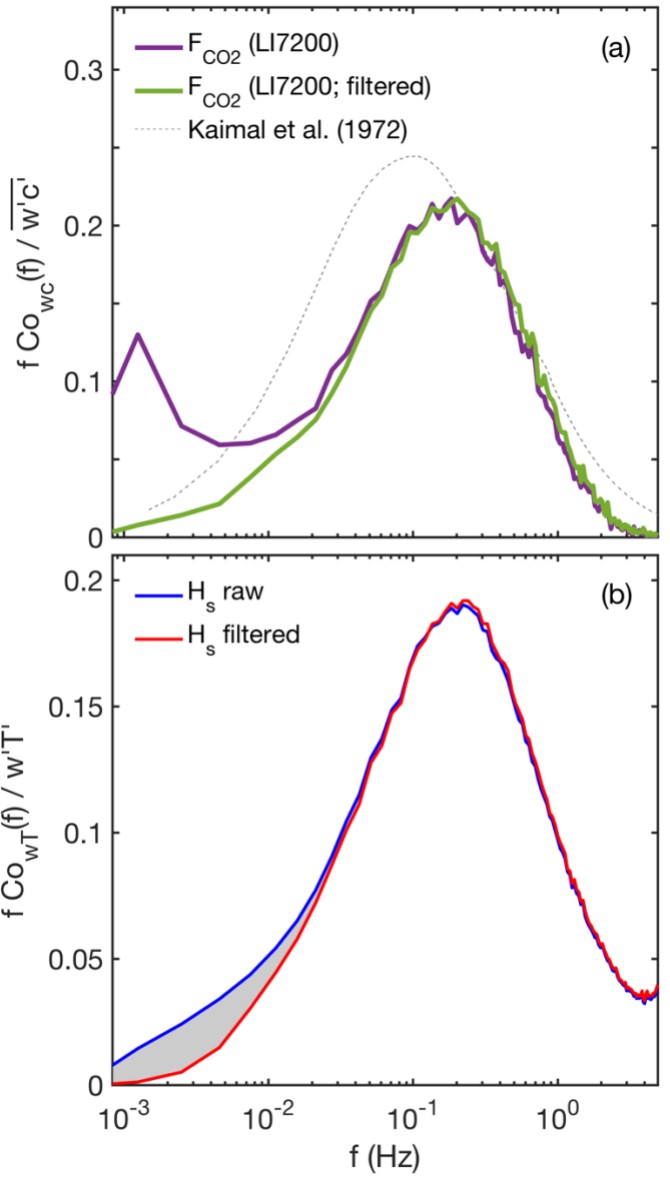

**Figure 5. Median normalized frequency-weighted cospectra for $F_{CO2}$ (a) and $H_S$ (b). In (a) the purple line represents the cospectrum calculated from the uncorrected LI-7200 $CO_2$ mixing ratio. The green curve represents the cospectrum calculated from the high-pass filtered LI-7200 $CO_2$ mixing ratio. The dashed gray line represents the theoretical scalar cospectra from Kaimal et al. (1972). In (b) the blue curve represents the cospectrum for $H_S$ calculated from an unfiltered air temperature measurement. The red curve represents the cospectrum for $H_S$ calculated from air temperature passed through the same high-pass filter applied to the $CO_2$ mixing ratio in the $F_{CO2}$ calculation (i.e., 1st order Butterworth filter with 0.005 Hz cutoff). The area under the shaded region between the two curves represents the median loss of real low frequency flux due to filtering.**





**Figure 6. Meteorological conditions from May to September 2017 shown as 3-hour averages for: (a) Incoming solar radiation (W m$^{-2}$), (b) air temperature (°C), (c); relative humidity (%), (d) atmospheric pressure (kPa), (e) wind speed (m s$^{-1}$), (f) wind direction (°), (g) $CO_2$ mixing ratio (ppm), and (h) water vapor concentration (ppt).**





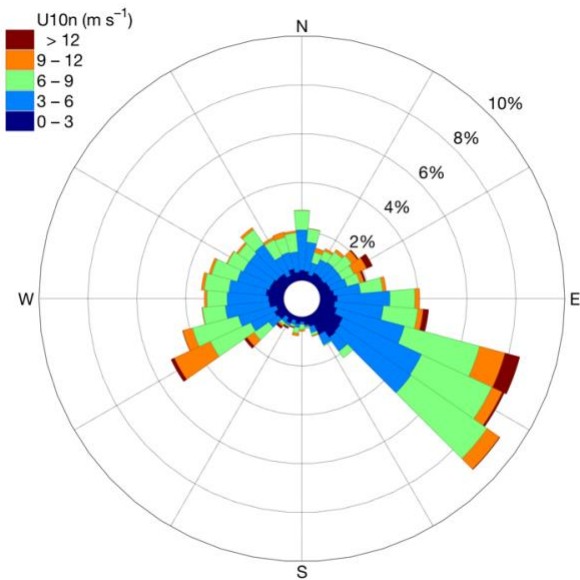

**Figure 7. Wind rose for May to September 2017 shown with 10° wind direction bins. Color represents 10 m neutral wind speed and the size of the bars indicates the frequency with which they occur.**

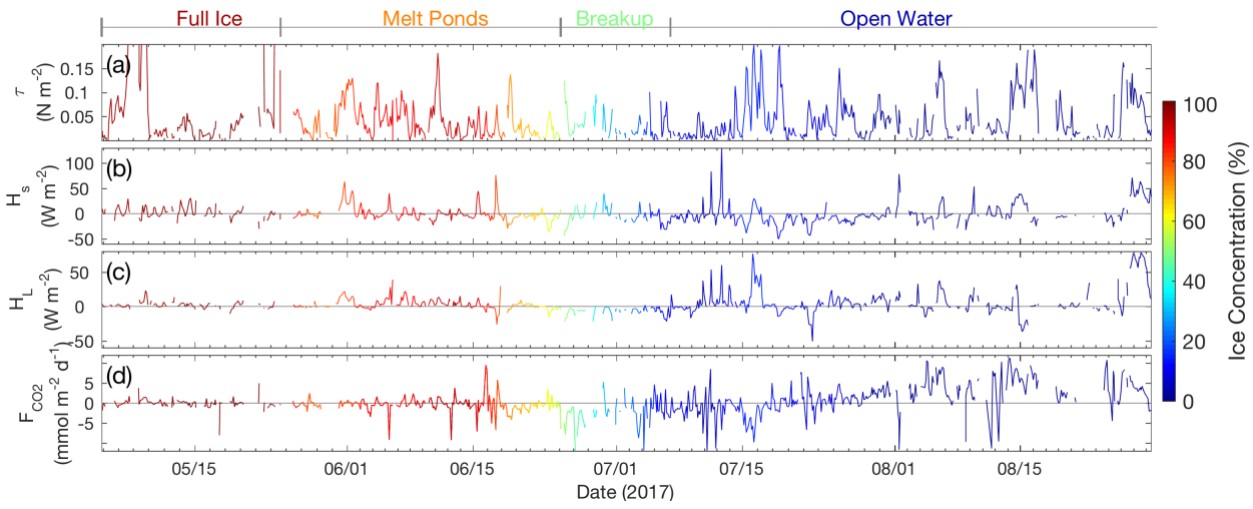

**Figure 8. Three-hour averages of measured fluxes, following the meteorological conventions (negative fluxes indicate transport towards the surface): (a) momentum flux (N m⁻²), (b) sensible heat flux (W m⁻²), (c) latent heat flux (W m⁻²), (d) CO₂ flux (mmol m⁻² d⁻¹), measured by the closed-path IRGA. Ice concentration from AMSR2 ice product is shown by color, with red representing full ice cover and blue representing open water. Demarcations (determined from satellite and in situ images) of ice regimes (full**

10 **ice, melt ponds, breakup, and open water) are shown on top.**




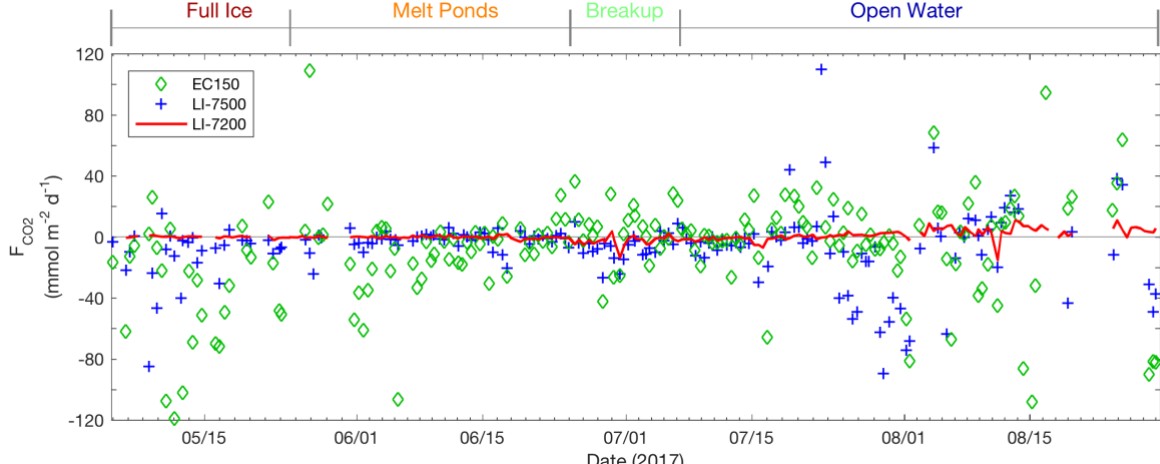

**Figure 9. Twelve-hour average CO₂ fluxes calculated using EC150 (green diamonds), LI-7500 (blue pluses), and dried LI-7200 (red line). Demarcations (determined from satellite and in situ images) of ice regimes (full ice, melt ponds, breakup, and open water) are shown on top.**




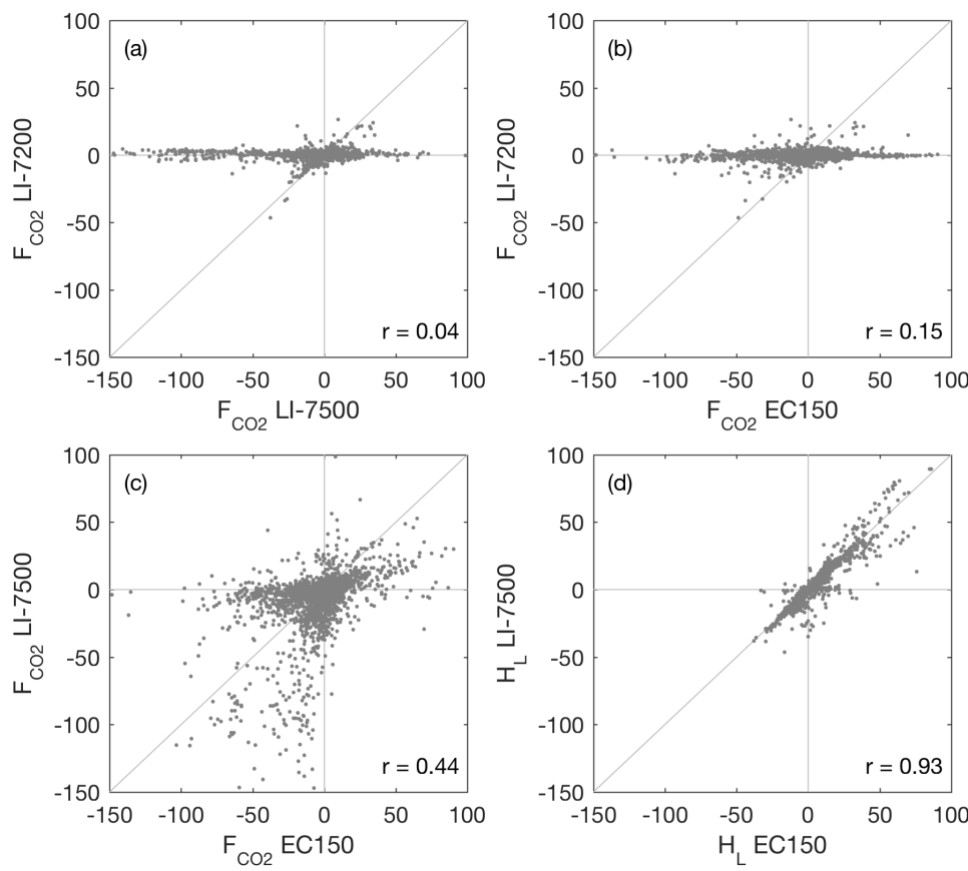

**Figure 10. Comparisons of F$_{CO2}$ and H$_L$ from the three IRGAs: (a) shows F$_{CO2}$ (LI-7200) vs. F$_{CO2}$ (LI-7500), (b) shows F$_{CO2}$ (LI-7200) vs. F$_{CO2}$ (EC150), (c) shows F$_{CO2}$ (LI-7500) vs. F$_{CO2}$ (EC150), and (d) shows H$_L$ (LI-7500) vs. H$_L$ (EC150). Because the sample air to the LI-7200 was dried comparisons to H$_L$ (LI-7200) were omitted.**

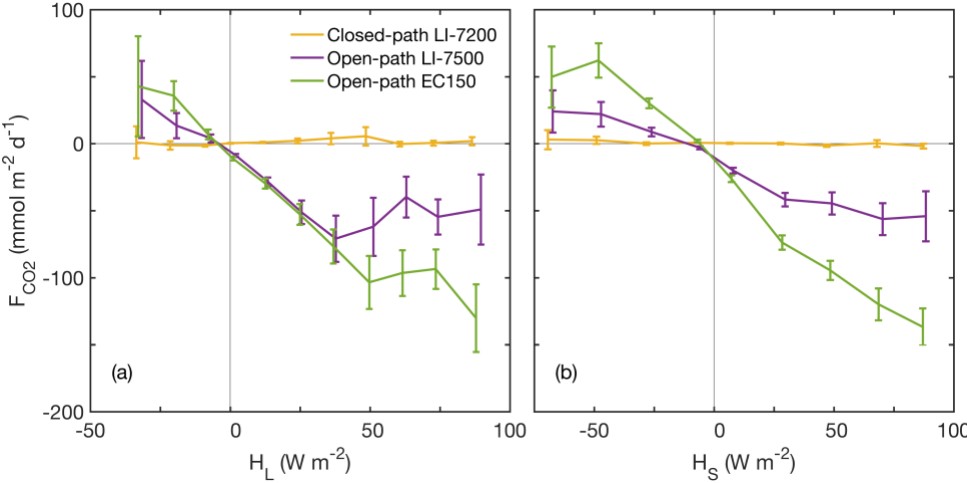

**Figure 11. Bin averages of the relationships between F$_{CO2}$ calculated from all three IRGAs and (a) H$_L$ (LI-7500) and (b) H$_S$.**





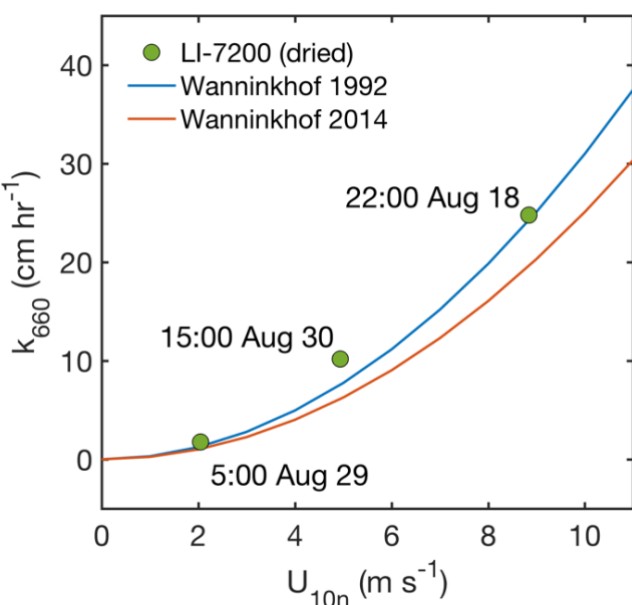

**Figure 12. Gas transfer velocity ($k_{660}$) plotted against 10 m neutral wind speed for three periods in which the RV *Martin Bergmann* measured pCO$_{2w}$ within 3 km of the tower and in which the magnitude of ΔpCO$_2$ (i.e., pCO$_{2w}$ − pCO$_{2atm}$) was greater than 20 µatm. Parameterizations of Wanninkhof (1992) and Wanninkhof (2014) shown as red and blue lines, respectively.**

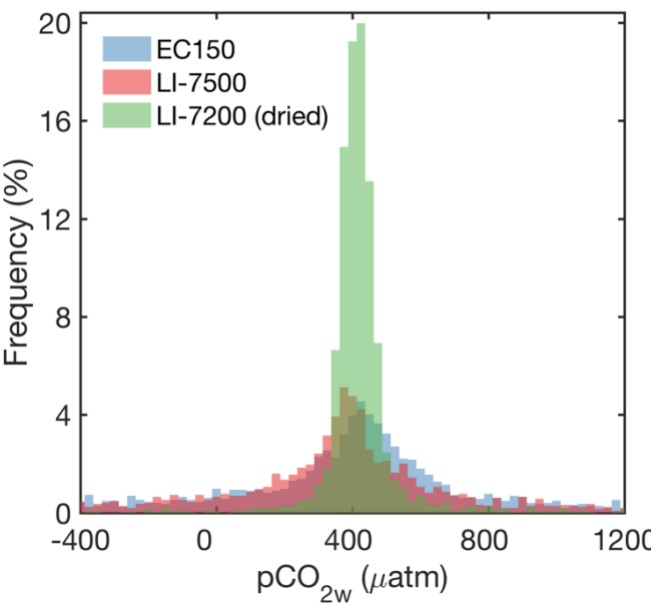

**Figure 13. Histograms showing the normalized frequency (%) of pCO$_{2w}$ calculated using F$_{CO2}$ from the EC150, LI-7500, and LI-7200. The limits in the x-axis were truncated at −400 and 1200 for better visualization. However, roughly a quarter of pCO$_{2w}$ values from both the EC150 and LI-7500 extend beyond these limits.**