# Peer review of "Dried, closed-path eddy covariance method for measuring carbon dioxide flux over sea ice"

_Atmospheric Measurement Techniques, 2018_

## Referee Comment (RC1) · Anonymous Referee #2 · 29 Sep 2018

I checked reviewer responses "ver.1 "File Upload (23 Jul 2018) by Brian Butterworth". Authors respond all my review comments and I agree with their all comments.
* * *

---

## Referee Comment (RC2) · J. Prytherch (Referee) · 3 Oct 2018

Review

This paper describes a system for measurement of air-sea and air-ice CO2 flux using the eddy covariance method and dried, closed-path infrared gas analyser instrumentation. The novel aspects are the successful deployment of such instrumentation in a particularly challenging environment (The Canadian Arctic) and the technical challenges overcome in doing so. The authors also present convincing evidence that the flux measurements from their system are plausible, and that the closed-path system performs much better than the different instruments (open-path infrared gas analysers)

[Figure]

previously used in such environments.

The paper is well written, well structured and thorough. I have a few questions and comments for the authors and some minor corrections. With these addressed, I am very happy to recommend this manuscript for publication in Atmospheric Measurement Techniques.

Comments

Section 2.1. and later. I felt like a little more discussion of the flux footprint might be useful. The far reaches of the footprint certainly seem likely to be over water (e..g the Kljun et al., 2015 model suggests for a 10 m tower the footprint doesn't extend further than 1km). However there can be significant influence from land close to the measurement site (20-100m). On page 8, line 11, you state that winds were selected within 150 deg of the front to reduce flow distortion from the mast and instruments but also because they included land in their footprint. From the photos and video provided, it appears that a far greater wind sector is potentially affected by the island. As the discussion on page 9 and fig 7. makes clear, most of the winds come from the southeast or southwest. It is also stated that the front of the tower facing the water is North. This at least suggests many of the measurement periods had winds blowing over a substantial part of the island. From the information given it is hard to determine how significant this might be. Further discussion, and a more detailed map showing the island itself, and the position and facing of the mast, would be useful additions. If it is the case that for many of the measurements, significant land (ie greater than say 20m) is within the footprint, then an estimate of this influence on the measurements should be made, using for example the footprint model above.

Section 2.2.1. Did the authors have any issues with ice formation, either on the sonic or the open path gas analysers? The model of sonic used does not appear to have any heating element. My experience is that for sonics, affected periods are straightforward to identify but this is not always the case for (open path) IRGAs. Did the authors use

any particular method to identify icing periods?

Section 2.2.2. Newer versions of the Li-7500 have a 'cold weather' mode, designed to reduce some of the biases apparent in these conditions. Did the model used in this study have such a mode and was it used or tested?

Section 3.1 and Figure 6. Humidity units are given in ppt. Presumably this is parts per thousand. My preference would be not to use this to avoid confusion with parts per trillion.

Section 4.4. The authors note that high spatial and temporal variability in the $pCO_{2w}$ measurements. Is this a potential explanation for the low frequency flux signal observed (ie 2.3.1), either from the temporally varying $pCO_{2w}$ itself, or from varying wind directions and the spatial variation in $pCO_{2w}$?

Section 4.4, page 13, line 13. I think the statement that open-path IRGAs are not capable of providing accurate $FCO_2$ measurements in the marine environment is somewhat overstated. As shown in Blomquist et al., 2014, the open path 7500 instrument has a lower sensitivity/signal to noise than the closed path 7200, and particularly the dried closed path instrument. If the flux is large enough (ie a large delta $pCO_2$) then an open path instrument can work well. When the flux is small, they perform badly. I think this is not necessarily a marine issue (e.g. Landwehr et al., 2014 showed that error in open path measurements did not seem to depend on the presence of hygroscopic particles (salt) as previously thought), though of course marine, and especially sea ice environments typically have much lower $CO_2$ fluxes than found on land. The sentence here should be slightly changed, perhaps to 'measurements in the relatively low delta $PCO_2$ conditions typically found in the marine environment'.

Section 4.6. The limit of 40 uatm for delta$pCO_2$ has been widely used historically, but I think it can be misleading. It was commonly used as a limit on ship-based studies with additional motion-based sources of noise. It may also be more appropriate for different instrumentation than that used in this study. The author's themselves use a limit of 20

uatm elsewhere (Figure 12) so maybe just omit the value here.

Corrections

Page 9 line 8. typo, change to 'snow-covered sea ice'.

Page 9 line 16. Not clear which period you are referring to. I think to the whole period presented in the paper, so perhaps change to 'through the study period'.
* * *

---

## Author Comment (AC1) · 17 Oct 2018

**Response to Reviewer 2**

Blue text:
**Comment:**  Referee comment. Page and line numbers relate to the original submission.

Black text:
**Response:**  Our comments to the reviewers and the public
**Action:**  What we have done to change this in the text of the article. Page and line numbers relate to the edited manuscript.

**Interactive Discussion**

**Comment:**  I checked reviewer responses "ver.1 "File Upload (23 Jul 2018) by Brian Butterworth". Authors respond all my review comments and I agree with their all comments.

**Response:**  We would like to thank the reviewer for the thoughtful comments. As a note to readers, the reviewer submitted comments in the initial submission's access review stage. Therefore, they were not posted in the interactive discussion online. We have posted those comments and our responses below.

**Access Review**

**General Comment:**  This manuscript presents the evaluation of the difference between enclosure and eddy covariance methods for CO2 flux measurements over sea ice. This manuscript provides new aspect to understanding the sea ice contribution to the CO2 budget to the atmosphere. On the other hand, this paper denied the data collected so far by eddy covariance in sea ice area. Can we use previous data or do you have idea to correct the vapor effect on CO2 concentration measurement for previous data (then CO2 flux data)? It is most important. You indicated that eddy covariance CO2 flux became similar to that for enclosure values and you said it is good. It seems that enclosure values are real (true) CO2 flux values. My impression is that enclosure method evaluated the underestimate CO2 flux over sea ice because this method does not consider the wind effect. Sea ice environment is basically high wind and there is no barrier to reduce the wind (not likely tree in the forest).

**Response:**  Without a simultaneous *undried*, closed-path LI7200 we do not have an adequate method for quantifying the water vapor impact to correct previous studies.

Because we did not collect enclosure measurements we are not claiming that our EC measurements are an exact match to enclosures, or that enclosures represent real/true values. This data simply highlights that using the best practices of EC (dried, closed-path) reduces sea ice flux measurements several orders of magnitude from suspiciously high open-path EC measurements. This moves sea ice EC measurements into a range that is more theoretically plausible. Future studies comparing the two methods simultaneously may provide insight into the degree to which enclosure method is influenced by its previously-documented shortcomings.

**Action:**   We felt that this comment showed we did not adequately emphasize the advantages of EC over enclosures.

Therefore, we changed Page 2, Line 22:
"Additionally, the measurements are spatially limited to the region enclosed by the chamber, making it challenging to investigate ecosystem-level questions."
To read:
"Additionally, the measurements are spatially and temporally limited. Measurements are confined to the region enclosed by the chamber (cm scale), making it challenging to accurately measure fluxes over whole ecosystems (m to km scale), which typically contain heterogeneity on scales larger than the footprint of the chamber. Additionally, long-term measurements are not feasible for enclosures due to the fact that they alter the underlying environment and the degree of manual intervention they require."

**Comment:**  Page 1, lines 1-2, Title: You should appeal your technical advancement with respect to the traditional method for CO2 flux over sea ice. For example, "The dried, closed-path EC method for CO2 flux measurement over sea ice".

**Response:**  We agree that this would improve the title.

**Action:**   Title now reads:
*Dried, closed-path eddy covariance method for measuring carbon dioxide flux over sea ice*

**Comment:**  Page 2, Line 3: Takahashi et al (2012) is focused on the Southern Ocean. Therefore, 2009 may better.

**Response:**  Okay.

**Action:**   Changed in line citation (Page 2, Line 3). Updated references to reflect the change.

**Comment:**  Page 2, Line 7: "CO2 fluxes over sea ice are small". As compared to what? If we check ice eddy covariance data, it is very high magnitude.

**Response:** $CO_2$ fluxes over sea ice are small compared to terrestrial fluxes and often small (though not always) compared to open ocean fluxes. The fact that sea ice EC has previously been measured at high magnitudes is addressed in this paper. We believe our position is clear that these measurements are not believable based on the literature over the past 5-10 years.

**Action:** On Page 2, Line 8 we changed "Generally, $CO_2$ fluxes over sea ice are small, but there are…" to read "Compared to terrestrial environments $CO_2$ fluxes over sea ice are small. However, there are…".

**Comment:** Page 2, Lines 22-23: "Ecosystem level question" I cannot understand. Enclosure method can also detect the ecosystem response.

**Response:** We refer reviewer to the following paragraph in Miller et al. (2015): "The greatest advantages and disadvantages to using enclosure methods are both due to spatial variability. Chamber enclosures only integrate the signal from the area they cover (generally, a few hundred $cm^2$); if the exchange is governed by factors that vary on larger horizontal scales (*i.e.*, the thickness and wetness of the snow cover, melt ponds, leads, under-ice hydrology, *etc.*), a prohibitive number of individual chamber measurements over a large area may be required to estimate the flux accurately (section 2.1). On the other hand, the method is ideal for studying specific, small-scale processes influencing variations in the flux (*i.e.*, brine channel distributions, ice algae respiration, *etc.*), and enclosure methods are the only technique available to determine fluxes on the same scale as most sea-ice biogeochemical measurements. In contrast, the micrometeorological techniques (sections 5.1.2–5.1.6) cover areas several orders of magnitude larger than chambers, integrating fluxes from different ice types and any open water in the footprint; micrometeorological results can, therefore, be difficult to interpret over heterogeneous surfaces."

**Action:** This comment was addressed with the action taken to the first comment (see above).

Additionally, on Page 15, Line 14 we changed "ecosystem-level processes" to "ecosystem-scale processes."

**Comment:** Page 2, Line 31: Why is it good agreement on terrestrial environment? No water vapor? We expected that in the polar area, water vapor amount is basically small.

**Response:** We can't say for sure why the terrestrial studies showed better agreement. But we believe that it stems from the larger terrestrial $CO_2$ fluxes resulting in a reduced impact (proportionally) of bias from interfering variables (e.g., water vapor cross talk) in the EC measurement.

The interference of $H_2O$ on the $CO_2$ concentration (in LI7200) appears to be greatest when the $CO_2$ fluxes are small. Over sea ice the $CO_2$ fluxes are very small compared to terrestrial

environments. Therefore, even small water vapor fluxes (when not preconditioned out of the sample air) cause a relatively large impact on the $CO_2$ measurement.

**Action:** Because we can't elaborate on the quality of the terrestrial flux measurements, commenting on them here appears to be a distraction. On Page 3, Line 1 we changed: "While the enclosure and EC methods have shown good agreement in terrestrial environments (Laine et al., 2006; Wang et al., 2013), they have not shown good agreement over sea ice…" to "The enclosure and EC methods have not shown good agreement over sea ice…"

We removed the associated citations from the references.

**Comment:** Page 4, Lines 26-: Why you did not prepare both dry/not dry LI-7200. In order to check the effect of dry or not dry air, same analyzer should use.

**Response:** We agree that this proposed setup could have yielded some additional information. We did not do it due to cost. Funds were only available to purchase one LI7200.

**Action:** No action taken.

**Comment:** Page 9, Lines 18-19: CO2 mixing ratio should indicated as ppmv?

**Response:** Yes, this is umol/mol.

**Action:** We have changed it to 'ppm' in the following places (Page 9, Lines 22-23 and Figure 6 caption). Figure 6 was updated so the ylabel for subplot g now reads 'ppm'.

**Comment:** Page 12, Line 1, 4.4 gas transfer velocity: You compared for open ocean data. I am not sure this is suitable because this paper focused on the $CO_2$ flux measurement over sea ice. Therefore, you should do this for ice pCO2. For open water condition, we do not need discussion.

**Response:** We believe this is an important section. The premise of this paper is that we don't really know what fluxes over sea ice should be, since enclosures have problems, and all previous EC attempts have had problems. The only way to check if our system is working is to compare it to something we do know, and that's the open ocean.

**Action:** No action taken.

**Comment:** Page 13, Lines 16-24: I cannot understand. You did not use zero air Nafion in this study. And, I think that H2O values become same for sometimes. Without a dry air counter flow, how to drying? Therefore, as indicated above, you need to check dry/not dry LI-7200 in order to clear up the effect of dry/not dry air to flux. Or you should remove/attach the desiccant for LI-

**Response:** Just to clarify, when the desiccant is active it fully eliminates water vapor going to the Nafion *counterflow*. But, the Nafion does not completely remove water vapor from the *sample* air. It reduces the magnitude of water vapor in the sample air and (this is the critical point) reduces fluctuations in water vapor. The important part with respect to the Nafion providing benefits even when the desiccant's capacity was exhausted, is that the Nafion still provides a smoothing effect on the water vapor fluctuations, even when not providing any reduction in the mean concentration of water vapor in the sample air. In this scenario, a spike of moist air will still exchange $H_2O$ with a counterflow that is at mean ambient humidity. Because the EC calculation relies on the fluctuations and not the means this smoothing of the water vapor fluctuations still provides benefits. Tubing itself (with no Nafion) does provide some smoothing to water vapor fluctuations as water vapor is 'stickier' than $CO_2$. But previous results from an undried LI7200 (Butterworth and Miller 2016b) showed this did not fully remove the influence of water vapor on the $CO_2$ flux, instead showing up as spurious low frequency contribution to the flux, which was visible in the flux cospectra. In this study, when the desiccant was exhausted, we did not see contribution in the cospectra suggestive of water vapor interference, suggesting that it is the Nafion that is helping.

A simultaneous measurement of undried LI7200 would likely provide some additional information and will be considered in the future if funding allows.

**Action:** On Page 13, Line 29 added:
This makes sense because spikes of moister or drier air will still exchange $H_2O$ with a counterflow that is at mean ambient humidity. Without a parallel, undried LI7200 it impossible to quantify how much of the smoothing is from the Nafion compared to the natural 'stickiness' of $H_2O$ on tube walls. However, the impact of $H_2O$ smoothing from tube walls alone was tested in Butterworth and Miller (2016b) with an undried LI7200. It was found that tubing did not fully remove the influence of water vapor on the $CO_2$ flux, and instead showed up as spurious low frequency contribution to the flux, which was visible in the flux cospectra. In this study, when the desiccant was exhausted, the cospectra did not indicate interference from water vapor, suggesting that that the Nafion played a critical role.

On Page 14, Line 4 changed "(particularly ones with power constraints)" to "(i.e., a system could be designed that uses a Nafion and counterflow, but without a dry air source)".

**Comment:** Page 13, Line 25: For future work, do you have idea to correct the vapor effect on CO2 concentration measurement for previous EC data (then CO2 flux data)? It is most important.

**Response:** Previous studies have compared dried and undried LI7200 in parallel (Landwehr et al. 2014, Blomquist et al. 2014, Butterworth and Miller 2016b). Quantifying a relationship for correcting $CO_2$ flux bias from water vapor in undried, closed-path systems does not appear to be straightforward due to large degree of scatter and inconsistent bias across different studies and across different IRGAs in the same study. Such work would constitute a full paper in and of itself and cannot be addressed here. It is something we plan to investigate in the future.

---

## Author Comment (AC2) · 17 Oct 2018

**Response to Reviewer 1**

Blue text:
**Comment:**     Referee comment. Page and line numbers relate to the original submission.

Black text:
**Response:**     Our comments to the reviewers and the public
**Action:**     What we have done to change this in the text of the article. Page and line numbers
relate to the edited manuscript.
* * *
**Review:**  This paper describes a system for measurement of air-sea and air-ice CO2 flux using the eddy covariance method and dried, closed-path infrared gas analyser instrumentation. The novel aspects are the successful deployment of such instrumentation in a particularly challenging environment (The Canadian Arctic) and the technical challenges overcome in doing so. The authors also present convincing evidence that the flux measurements from their system are plausible, and that the closed-path system performs much better than the different instruments (open-path infrared gas analysers) previously used in such environments.

The paper is well written, well structured and thorough. I have a few questions and comments for the authors and some minor corrections. With these addressed, I am very happy to recommend this manuscript for publication in Atmospheric Measurement Techniques.

**Response:**     We'd like to thank Dr. John Prytherch for taking the time and providing a thoughtful review.

**Comment:**     Section 2.1. and later. I felt like a little more discussion of the flux footprint might be useful. The far reaches of the footprint certainly seem likely to be over water (e..g the Kljun et al., 2015 model suggests for a 10 m tower the footprint doesn't extend further than 1km). However there can be significant influence from land close to the measurement site (20-100m). On page 8, line 11, you state that winds were selected within 150 deg of the front to reduce flow distortion from the mast and instruments but also because they included land in their footprint. From the photos and video provided, it appears that a far greater wind sector is potentially affected by the island. As the discussion on page 9 and fig 7. makes clear, most of the winds come from the southeast or southwest. It is also stated that the front of the tower facing the water is North. This at least suggests many of the measurement periods had winds blowing over a substantial part of the island. From the information given it is hard to determine how significant this might be. Further discussion, and a more detailed map showing the island itself, and the

position and facing of the mast, would be useful additions. If it is the case that for many of the measurements, significant land (ie greater than say 20m) is within the footprint, then an estimate of this influence on the measurements should be made, using for example the footprint model above.

**Response:**     We looked into this in greater detail. Our original decision to keep such a wide swath was due to the fact that we appeared to get fluxes from these back-hemisphere angles (i.e., believable magnitudes with reasonable-looking cospectra). As per your suggestion we ran the Kljun et al. (2015) model using the mean meteorological conditions from every 5 degree wind sector. We found that for the back-hemisphere angles in question 44% of the flux was coming from the island. Though there were many periods in which that number was much lower. For example, when we ran the same analysis, but input meteorological conditions representative of the $75^{th}$ percentiles (instead of means) we found this influence dropped to 23%. Overall, we believe that there may still be useable data from these angles. Though, without further data (specifically a time series of $pCO_{2w}$) we cannot properly address it now. Fortunately, these back-hemisphere wind directions were relatively uncommon. Since the tower was pointed northwest (330°, as stated in the article), the common southeasterly winds were already discarded. We did rerun our plots and statistics using a smaller angle limit (−90° to 90°) and found that it would change the conclusions of the paper in no discernable way. Therefore, we report on this finding, but left the wind direction limits as they were. And we left the possibility open to applying a correction factor down the road when we can test it with a $pCO_{2w}$ dataset.

**Action:**        On page 8, line 10 we changed:

"Quality control criteria were used to select intervals that passed the underlying assumptions of eddy covariance. First, wind directions of −150° to 150° (relative to the anemometer) were selected to eliminate winds from aft that were affected by flow distortion from the instruments and tower frame. Winds from aft were also unwanted as they included the island land surface in their flux footprint. A second quality control criterion selected for intervals that exhibited stationarity following

$$RN_{cov} = \left| \frac{\overline{(w'x')}_5 - \overline{(w'x')}_{20}}{\overline{(w'x')}_{20}} \right| \leq 0.3, \tag{8}$$

where $\overline{(w'x')}_5$ represents the mean of the four 5-minute turbulent flux subintervals and $\overline{(w'x')}_{20}$ represents the turbulent flux of the whole 20-minute interval (Blomquist et al., 2014). The purpose of this criterion is to identify and remove intervals in which large scale phenomena (e.g., mesoscale motions), outside the frequency range of turbulent fluxes, are contributing to the measured flux."

To read:

"Quality control criteria were used to select intervals that passed the underlying assumptions of eddy covariance. First, intervals were selected that exhibited stationarity, following

$$RN_{cov} = \left| \frac{\overline{(w'x')}_5 - \overline{(w'x')}_{20}}{\overline{(w'x')}_{20}} \right| \leq 0.3, \tag{8}$$

where $\overline{(w'x')}_5$ represents the mean of the four 5-minute turbulent flux subintervals and $\overline{(w'x')}_{20}$ represents the turbulent flux of the whole 20-minute interval (Blomquist et al., 2014). The purpose of this criterion is to identify and remove intervals in which large scale phenomena (e.g., mesoscale motions), outside the frequency range of turbulent fluxes, are contributing to the measured flux.

A second quality control criterion selected for wind directions of −150° to 150° (relative to the anemometer) to eliminate winds from aft that were affected by flow distortion from the instruments and tower frame. The size and shape of the island (0.2 km wide and extending 0.5 km behind the tower) meant the remaining wind directions from the back hemisphere had some degree of island contributing to their flux footprint. To estimate the impact of the island on the flux measurements we ran the flux footprint model of Kljun et al. (2015). Using the mean meteorological conditions from each 5° wind sector it was found that on average the island accounted for 5% of the footprint for wind directions from the front hemisphere (−90° to 90°). From the back wind sectors (−150° to −90° and 90° to 150°) 44% of the flux footprint was represented by the island. However, because extent of the footprint varied with meteorological conditions there were many periods where these wind directions saw minimal influence from the island. The island is bare rock, which means that it should not act as either a source or a sink for $CO_2$. This means that the magnitudes of $CO_2$ fluxes from these back sectors are somewhat underestimated, by a factor that most likely scales with the portion of the footprint that falls on the island. Future work is planned to collect field data (i.e., coincident upwind $pCO_{2w}$ data for varying wind directions) to determine if a linear scaling factor could be used to correct flux magnitudes for these wind sectors."

We added the following reference:

Kljun, N., Calanca, P., Rotach, M. W. and Schmid, H. P.: A simple two-dimensional parameterization for Flux Footprint Prediction ( FFP ), Geosci. Model Dev., 8, 3695–3713, doi:10.5194/gmd-8-3695-2015, 2015.

**Comment:** Section 2.2.1. Did the authors have any issues with ice formation, either on the sonic or the open path gas analysers? The model of sonic used does not appear to have any heating element. My experience is that for sonics, affected periods are straightforward to identify but this is not always the case for (open path) IRGAs. Did the authors use any particular method to identify icing periods?

**Response:** For this study period we did not have issues with ice formation. To assess icing we mounted the GoPro camera so that it included the sonic and the LI7500 in its view. Unfortunately, when there is icing the camera also ices over, though it is still an indication that equipment is icing at the top of the mast. Our other method for identifying periods of icing was to use the CSAT3 diagnostic.

**Comment:** Section 2.2.2. Newer versions of the Li-7500 have a 'cold weather' mode, designed to reduce some of the biases apparent in these conditions. Did the model used in this study have such a mode and was it used or tested?

**Response:** Yes, our LI-7500 does have a 'cold weather' mode. It was set to this mode.

**Comment:** Section 3.1 and Figure 6. Humidity units are given in ppt. Presumably this is parts per thousand. My preference would be not to use this to avoid confusion with parts per trillion.

**Response:** We are most comfortable with ppt.

**Action:** To avoid confusion we define ppt when it is first used. Page 10, Line 2 now reads: "mean of 3.3 parts per thousand (ppt)"

**Comment:** Section 4.4. The authors note that high spatial and temporal variability in the pCO2w measurements. Is this a potential explanation for the low frequency flux signal observed (ie 2.3.1), either from the temporally varying pCO2w itself, or from varying wind directions and the spatial variation in pCO2w?

**Response:** It's an interesting question. We don't believe the low frequency contribution is from spatially or temporally varying $pCO_{2w}$ because we see it most strongly during the full sea ice coverage periods, when there is likely very small spatial and temporal $pCO_{2w}$ variation.

**Comment:** Section 4.4, page 13, line 13. I think the statement that open-path IRGAs are not capable of providing accurate FCO2 measurements in the marine environment is somewhat overstated. As shown in Blomquist et al., 2014, the open path 7500 instrument has a lower sensitivity/signal to noise than the closed path 7200, and particularly the dried closed path instrument. If the flux is large enough (ie a large delta pCO2) then an open path instrument can work well. When the flux is small, they perform badly. I think this is not necessarily a marine issue (e.g. Landwehr et al., 2014 showed that error in open path measurements did not seem to depend on the presence of hygroscopic particles (salt) as previously thought), though of course marine, and especially sea ice environments typically have much lower CO2 fluxes than found on land. The sentence here should be slightly changed, perhaps to 'measurements in the relatively low delta PCO2 conditions typically found in the marine environment'.

**Response:** Agreed, if the $dpCO_2$ was large enough the open-path would work.

**Action:** We made the suggested change (see page 13, line 26).

**Comment:** Section 4.6. The limit of 40 uatm for deltapCO2 has been widely used historically, but I think it can be misleading. It was commonly used as a limit on ship-based studies with additional motion-based sources of noise. It may also be more appropriate for different instrumentation than that used in this study. The author's themselves use a limit of 20 uatm elsewhere (Figure 12) so maybe just omit the value here.

**Response:** Agreed, no need to include it.

**Action:** We omitted it.

**Corrections:**

Page 9 line 8. typo, change to 'snow-covered sea ice'.

**Action:** Changed.

Page 9 line 16. Not clear which period you are referring to. I think to the whole period presented in the paper, so perhaps change to 'through the study period'.

**Action:** Changed.

[revised manuscript text omitted]